# Enhancing Diffusion Posterior Sampling for Inverse Problems by Integrating Crafted Measurements

## Abstract

Diffusion models have emerged as a powerful foundation model for visual generation. With an appropriate sampling process, it can effectively serve as a generative prior to solve general inverse problems. Current posterior sampling based methods take the measurement (i.e., degraded image sample) into the posterior sampling to infer the distribution of the target data (i.e., clean image sample). However, in this manner, we show that high-frequency information can be prematurely introduced during the early stages, which could induce larger posterior estimate errors during the restoration sampling. To address this issue, we first reveal that forming the log posterior gradient with the noisy measurement ( i.e., samples from a diffusion forward process) instead of the clean one can benefit the reverse process. Consequently, we propose a novel diffusion posterior sampling method DPS-CM, which incorporates a Crafted Measurement (i.e., samples generated by a reverse denoising process, compared to random sampling with noise in standard methods) to form the posterior estimate. This integration aims to mitigate the misalignment with the diffusion prior caused by cumulative posterior estimate errors. Experimental results demonstrate that our approach significantly improves the overall capacity to solve general and noisy inverse problems, such as Gaussian deblurring, super-resolution, inpainting, nonlinear deblurring, and tasks with Poisson noise, relative to existing approaches.

## 1 Introduction

Diffusion models (Ho et al., 2020) have achieved remarkable generative performance on images (Amit et al., 2021; Baranchuk et al., 2021; Brempong et al., 2022), videos (Singer et al., 2022; Wu et al., 2023), audios (Popov et al., 2021; Yang et al., 2023a), natural language (Austin et al., 2021; Hoogeboom et al., 2021; Li et al., 2022) and molecular generation (Hoogeboom et al., 2022; Jing et al., 2022). Besides its strong modeling capacity for complex and high dimensional data, diffusion models have exhibited a strong generative prior to form the diffusion conditional sampling (Song et al., 2020) that can be harnessed for diffusion posterior sampling. In the context of noisy inverse problems, this sampling process effectively approximates precise data distributions from noisy and degraded measurements. Noisy inverse problems, such as super-resolution, inpainting, linear and nonlinear deblurring, are targeted to restore an unknown image $x$ from its noise corrupted measurement $y$ given the corresponding forward measurement operators $\mathcal{A}(\cdot) : \mathbb{R}^n \to \mathbb{R}^m$. Recently, Diffusion models have been extensively utilized for these tasks (Zhu et al., 2023; Song et al., 2022; Wang et al., 2022), offering a robust framework for reconstructing high-quality images from degraded measurements.

Current diffusion-based methods generally employ two distinct strategies to solve inverse problems. The first strategy is to train problem-specialized diffusion models (Saharia et al., 2022; Whang et al., 2022; Luo et al., 2023; Chan et al., 2023) given measurements and clean image pairs. In contrast, methods of the second strategy only capitalize on problem-agnostic pre-trained diffusion models to benefit the zero-shot diffusion restoration sampling by posterior estimate (Song et al., 2023a; Rout et al., 2024; Peng et al., 2024; Mardani et al., 2023) or enforcing data consistency (Chung et al., 2022b). In this paper, we concentrate on the second manner, posterior estimate for sampling, to solve noisy inverse problems universally.

Posterior estimate based methods enable controllable generations (Dhariwal & Nichol, 2021) via diffusion conditional reverse-time SDE (Song et al., 2020), such as classifier guidance (Dhariwal & Nichol, 2021), loss-guided diffusion (Song et al., 2023b). Applying Bayesian, the gradient of log posterior $\nabla_{\boldsymbol{x}_t} \log p_t (\boldsymbol{x}_t \mid \boldsymbol{y})$ can be easily adopted into conditional reverse-time SDE sampling as a log likelihood gradient term $\nabla_{\boldsymbol{x}_t} \log p (\boldsymbol{y} \mid \boldsymbol{x}_t)$ and an unconditional prior term $\nabla_{\boldsymbol{x}_t} \log p_t (\boldsymbol{x}_t)$. In the context of solving inverse problems, however, $p (\boldsymbol{y} \mid \boldsymbol{x}_t)$ is an intractable distribution due to the unclear dependency between the measurement $\boldsymbol{y}$ and the diffusion generation $\boldsymbol{x}_t$ at time $t$. To tackle this issue, existing posterior estimate methods for inverse problems, such as Diffusion Posterior Sampling (DPS (Chung et al., 2022a)), form the measurement model $p (\boldsymbol{y} \mid \hat{\boldsymbol{x}}_0)$ as the likelihood estimate, where $\hat{\boldsymbol{x}}_0$ is the denoising prediction given diffusion intermediate output $\boldsymbol{x}_t$. This process maps $\hat{\boldsymbol{x}}_0$ into the measurement $\boldsymbol{y}$'s space, which can be interpreted as a reconstruction loss guidance to push $\mathcal{A} (\hat{\boldsymbol{x}}_0)$ close to $\boldsymbol{y}$, while narrowing the gap between $\boldsymbol{x}_t$ and the clean image $\boldsymbol{x}$.

However, via empirical examples in Section 3.1, we show that solving inverse problems in the manner of diffusion posterior sampling follows a similar pattern of diffusion reverse process (Yang et al., 2023b), i.e., focusing on low-frequency recovery at first and posing increasing attention on high-frequency generation in the late stages. With such observation, the log posterior gradient estimate $\nabla_{\boldsymbol{x}_t} \log p (\boldsymbol{y} \mid \hat{\boldsymbol{x}}_0)$ in DPS with sharp measurement $\boldsymbol{y}$ will easily introduce abrupt high-frequency gradient signals for the subsequent step, which unfits the appropriate input pattern for the pre-trained model $\boldsymbol{s}_\theta (\boldsymbol{x}_t, t)$ during the early stages with large $t$. In fact, Song et al. (2023b) also note that DPS significantly miscalculates the scale of the guidance term with different variance levels which results in accumulated errors in posterior sampling. In Section 3.1, we have the similar observations that DPS amplifies posterior sampling errors. We also find that applying the likelihood estimate $p (\boldsymbol{y}_t \mid \hat{\boldsymbol{x}}_0)$ with randomly sampled **noisy measurement** $\boldsymbol{y}_t$ instead of the clean measurement $\boldsymbol{y}$ in DPS for each timestep $t$ leads to a smaller approximation error during the early stages, and thus benefits the restoration generation. Posterior approximation with **noisy measurement** $\boldsymbol{y}_t$ at timestep $t$, compared with the clean one, has the advantage that it adaptively matches the frequency pattern of the diffusion model's generation at the timestep $t$.

Therefore, with this insight, we propose the posterior approximation that leads to less high-frequency signal recovery during the early stages. Specifically, we propose the Diffusion Posterior Sampling with **Crafted Measurements (DPS-CM)**, which can introduce less biased posterior estimate by combining crafted measurement $\mathbf{y}_t$, **the intermediate generation of another diffusion reverse-time trajectory** $\{\mathbf{y}_t\}_{t=0}^T$ from posterior $p (\mathbf{y}_t \mid \boldsymbol{y})$. As $\{\mathbf{y}_t\}_{t=0}^T$ shares a similar frequency distribution pattern with the target generation trajectory $\{\boldsymbol{x}_t\}_{t=0}^T$, i.e., low-frequency recovery at first, the approximated log likelihood gradient $\nabla_{\boldsymbol{x}_t} \log p (\hat{\mathbf{y}}_0 \mid \hat{\boldsymbol{x}}_0)$ will bring in less high-frequency gradient signal and thus benefit the subsequent generations. Besides, leveraging **crafted measurements** $\mathbf{y}_t$ brings the lower bias compared with directly using randomly sampled **noisy measurement** $\boldsymbol{y}_t$. Our extensive experiments results on various noisy linear inverse problems, e.g. super-resolution, random masked/fixed box inpainting, Gaussian/Motion deblurring, and nonlinear inverse problems such as nonlinear deblurring, demonstrate that the proposed DPS-CM significantly outperforms existing unsupervised methods on both FFHQ (Karras et al., 2019) and ImageNet (Deng et al., 2009) datesets while keeping the algorithm simplicity.

## 2 BACKGROUND

### 2.1 DIFFUSION MODELS

Diffusion models (Ho et al., 2020; Song et al., 2020) comprise two processes: a forward noising process in which the noise is progressively injected into the sample and a reverse denoising process for generation. Specifically, the forward process of Denoising Diffusion Probabilistic Models (DDPM) (Ho et al., 2020) can be formulated by the variance preserving stochastic differential equation (VP-SDE) (Song et al., 2020):

$$d\boldsymbol{x} = -\frac{\beta_t}{2} \boldsymbol{x} dt + \sqrt{\beta_t} d\boldsymbol{w}, \quad t \in [0, T], \tag{1}$$

where $\{\beta_t\}_{t=0}^T$ is the designed noise schedule which is monotonically increased and $\boldsymbol{w}$ is the standard Wiener process. To generate samples of targeted data distribution from Gaussian noise

$x_T \sim \mathcal{N}(0, I)$, corresponding reverse SDE of Eq. 1 is formulated as:

$$dx = \left[ -\frac{\beta_t}{2} x - \beta_t \nabla_{x_t} \log p(x_t) \right] dt + \sqrt{\beta_t} d\bar{w}, \quad (2)$$

where $d\bar{w}$ is the reverse standard Wiener process and $\nabla_{x_t} \log p(x_t)$ is the score function of denoised sample at time $t$. Approximating this score function is the key to solving the reverse generative process, which can be done by training a parameterized model $s_\theta(x_t, t)$ by denoising score matching (Song & Ermon, 2019) expected on samples from the forward diffusion process:

$$x_t = \sqrt{\bar{\alpha}_t} x_0 + \sqrt{1 - \bar{\alpha}_t} z, \quad z \sim \mathcal{N}(0, I). \quad (3)$$

With trained score model $s_\theta$, applying Tweedie's formula (Efron, 2011; Dunn & Smyth, 2005), we are able to approximate the mean $\hat{x}_0$ of posterior $p(x_0 \mid x_t)$ for the above case (VP-SDE) as:

$$\hat{x}_0 := \mathbb{E}[x_0 \mid x_t] \simeq \frac{1}{\sqrt{\bar{\alpha}(t)}} (x_t + (1 - \bar{\alpha}(t)) s_\theta(x_t, t)). \quad (4)$$

## 2.2 SOLVING NOISY INVERSE PROBLEMS WITH DIFFUSION MODELS

Given a clean image $x_0$ and a forward measurement operator $\mathcal{A}(\cdot)$, the general form of the noisy inverse problem can be stated as:

$$y = \mathcal{A}(x_0) + \eta,$$

where an i.i.d. noise $\eta \sim \mathcal{N}(0, \sigma^2 I)$ is added in $\mathcal{A}(x_0)$ in the case of Gaussian noise to formulate the noisy inverse problem. Given measurements $y \in \mathbb{R}^m$ and a known forward measurement operator $\mathcal{A}(\cdot) : \mathbb{R}^n \to \mathbb{R}^m$, the target is to restore the unknown signal $x_0 \in \mathbb{R}^n$. Replacing unconditional score $\nabla_{x_t} \log p(x_t)$ in Eq. 2 with the conditional score $\nabla_{x_t} \log p(x_t \mid y)$ paves the way for solving inverse problems using diffusion models. Applying the Bayesian rule on the gradient of posterior $\nabla_{x_t} \log p(x_t \mid y)$, the reverse-time SDE of the posterior sampling is given by:

$$dx = \left[ -\frac{\beta_t}{2} x - \beta_t (\nabla_{x_t} \log p(x_t) + \nabla_{x_t} \log p(y \mid x_t)) \right] dt + \sqrt{\beta_t} d\bar{w}. \quad (5)$$

However, the gradient of the log likelihood $\nabla_{x_t} \log p(y \mid x_t)$ in Eq. 5 is intractable given $y = \mathcal{A}(x_0) + \eta$. DDRM (Kawar et al., 2022) addresses this issue by forming the conditional sampling in the spectral space without dealing with $\nabla_{x_t} \log p(y \mid x_t)$, but with the restriction to handle complex inverse problems due to the SVD computation efficiency. DPS (Chung et al., 2022a) provides a universal posterior approximation approach to deal with the intractable likelihood term $\nabla_{x_t} \log p(y \mid x_t)$. Given mean estimate $\hat{x}_0$ of posterior $\mathbb{E}[x_0 \mid x_t]$ by applying Eq.4, DPS assumes:

$$p(y \mid x_t) = \mathbb{E}_{x_0 \sim p(x_0 \mid x_t)} [p(y \mid x_0)]$$
$$\simeq p(y \mid \hat{x}_0). \quad (6)$$

As the production, they form the gradient of the posterior as:

$$\nabla_{x_t} \log p_t(x_t \mid y) \simeq s_{\theta^*}(x_t, t) - \rho \nabla_{x_t} \|y - \mathcal{A}(\hat{x}_0)\|_2^2.$$

Although DPS provides a straightforward and task-independent approximation, many existing works, such as Peng et al. (2024); Song et al. (2023b), point out this posterior estimate is biased.

# 3 METHOD: DIFFUSION POSTERIOR SAMPLING WITH CRAFTED MEASUREMENTS

We propose DPS-CM, which perform diffusion posterior sampling from $p(x_t \mid y_t)$ with **Crafted Measurement** $y_t$ belonging to another diffusion reverse trajectory $\{y_t\}_{t=0}^T$ instead of the vanilla input $y$. Specifically, applying Eq.4 on $x_t$ and $y_t$, we incorporate the tractable $p(\hat{y}_0 \mid \hat{x}_0)$ as the likelihood approximation of $p(y_t \mid x_t)$ to enable the posterior sampling from $p(x_t \mid y_t)$. DPS-CM can be interpreted as pushing the bond between two noisy approximation $\hat{y}_0$ and $\hat{x}_0$ as the measurement model $\hat{y}_0 = \mathcal{A}(\hat{x}_0)$. While $\hat{y}_0$ is progressively recovered to the input measurement $y$ via its diffusion denoising process, $\hat{x}_0$ is expected to develop into the target ground truth $x$ gradually during the synchronous reverse process with the measurement bond $\hat{y}_0 = \mathcal{A}(\hat{x}_0)$. In Section 3.1, we explore the advantage of posterior sampling utilizing noisy measurements $y_t$ sampled from the diffusion forward process over using the clean input $y$ in reducing intermediate generation errors in diffusion posterior sampling. In Section 3.2, we formulate the DPS-CM in detail.

## 3.1 DPS Amplify Posterior Sampling Errors

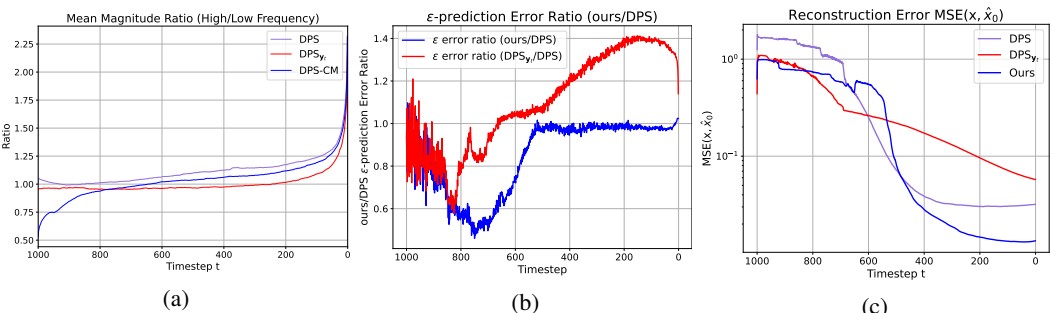

(a)                                     (b)                                     (c)

Figure 1: (a) mean magnitude ratio between high- and low-frequency area at timestep $t$; (b) $\epsilon$-prediction error ratio: $\frac{\text{DPS}_{y_t}}{\text{DPS}}$ and $\frac{\text{DPS-CM}}{\text{DPS}}$ at timestep $t$; (c) reconstruction error between target $\boldsymbol{x}$ and intermediate $\hat{\boldsymbol{x}}_0(\boldsymbol{x}_t)$ at timestep $t$. These results are achieved by performing Gaussian Deblurring on 10 FFHQ images.

Our claim is that solving inverse problems in the posterior sampling manner follows the similar frequency dynamics with the regular denoising process that it recovers the low-frequency (structural, domain-independent) components at first and the high-frequency (details, domain-dependent) components later. The posterior sampling from $p_t(\boldsymbol{x}_t \mid \boldsymbol{y})$ in DPS is inevitable adding high-frequency signals from the gradient of the approximate log likelihood $\nabla_{\boldsymbol{x}_t} \log p_t(\boldsymbol{y} \mid \hat{\boldsymbol{x}}_0)$ during the early stages due to the frequency and noise discrepancy between the sharp measurement $\boldsymbol{y}$ and intermediate noisy reconstruction $\hat{\boldsymbol{x}}_0$. It will create a distorted frequency dynamic which leads to the larger posterior sampling error.

To verify our assumption, we conduct frequency dynamic analysis of different posterior sampling methods for solving inverse problems in Fig.1a. Afterward, the posterior sampling error is illustrated via the visualization of the $\epsilon$-prediction error and intermediate reconstruction error dynamics in Fig.1b and Fig.1c respectively. We conduct the diffusion posterior sampling from $p_t(\boldsymbol{x}_t \mid \boldsymbol{y}_t)$ with noisy measurement $\text{DPS}_{\boldsymbol{y}_t}$ in comparison with DPS to illustrate the advantage of the likelihood approximation with less high-frequency elements.

**Diffusion posterior sampling with the noisy measurement $\text{DPS}_{\boldsymbol{y}_t}$.** Given the noisy measurement $\boldsymbol{y}_t$ constructed by the diffusion forward process starting from the clean measurement $\boldsymbol{y}$ as in Eq. 3, we replace $\boldsymbol{y}$ with $\boldsymbol{y}_t$ to form $\text{DPS}_{\boldsymbol{y}_t}$ with the posterior sampling from $p_t(\boldsymbol{x}_t \mid \boldsymbol{y}_t)$. The corresponding likelihood approximation $p_t(\boldsymbol{y}_t \mid \hat{\boldsymbol{x}}_0) \simeq p_t(\boldsymbol{y}_t \mid \boldsymbol{x}_t)$ involves $\boldsymbol{y}_t$ and $\boldsymbol{x}_t$ with the similar frequency pattern and noise level.

**Frequency dynamic of diffusion posterior sampling.** We verify our assumption on frequency dynamic of diffusion posterior sampling by transforming the gradient of log posterior (i.e., $\nabla_{\boldsymbol{x}_t} \log p(\boldsymbol{x}_t \mid \boldsymbol{y})$ in the case of DPS) into spectral space as the gradient of log posterior serving as the main update for each posterior sampling step in Eq.5. We split the low-frequency and high-frequency areas in the spectral space of the gradient with the low-high cut-off frequency of 32Hz and visualize the mean magnitude ratio between them of each $t$ shown in Fig.1a.

**$\epsilon$-prediction error and intermediate reconstruction error.** Since there exists no ground truth for the intermediate generation of diffusion posterior sampling for inverse problems, we measure the posterior sampling error implicitly by computing $\epsilon$-prediction error and intermediate reconstruction error at timestep $t$. Given intermediate $\boldsymbol{x}_t$ and the posterior mean $\hat{\boldsymbol{x}}_0$ by applying Eq.4, the $\epsilon$-prediction error is formulated as $\|\epsilon_\theta(\boldsymbol{x}_t', t) - \epsilon\|_2^2$, where $\boldsymbol{x}_t' = \sqrt{\bar{\alpha}_t}\hat{\boldsymbol{x}}_0 + \sqrt{1 - \bar{\alpha}_t}\epsilon, \epsilon \sim \mathcal{N}(\mathbf{0}, \boldsymbol{I})$ and $\epsilon_\theta(\boldsymbol{x}_t', t) = -\sqrt{1 - \bar{\alpha}_t}\boldsymbol{s}_\theta(\boldsymbol{x}_t', t)$ by applying Tweedie's formula. $\epsilon$-prediction error at timestep $t$ can reflect how accurately the posterior sampling recovers at $t$. We compare different posterior sampling with DPS by computing their ratio of $\epsilon$-prediction error at each timestep shown in Fig.1b, where a ratio below 1 indicates a smaller $\epsilon$-prediction error than DPS, and above 1 means a larger one. Besides, the reconstruction error formed as $\text{MSE}(\boldsymbol{x}, \hat{\boldsymbol{x}}_0)$ is shown in Fig.1c to evaluate how well the the intermediate reconstruction $\hat{\boldsymbol{x}}_0(\boldsymbol{x}_t)$ converges to the target $\boldsymbol{x}$ utilizing the mean squared error.

Figure 2: **Qualitative example of Gaussian Deblurring on FFHQ dataset including DPS$_{\boldsymbol{y}_t}$.** All measurements are corrupted by additional Gaussian noise with a variance of $\sigma_{\boldsymbol{y}} = 0.05$.

**Observations.** From Fig.1a, the frequency dynamic of both DPS's $\nabla_{\boldsymbol{x}_t} \log p\left(\boldsymbol{x}_t \mid \boldsymbol{y}\right)$ (purple) and DPS$_{\boldsymbol{y}_t}$'s $\nabla_{\boldsymbol{x}_t} \log p\left(\boldsymbol{x}_t \mid \boldsymbol{y}_t\right)$ (red) are focusing on low-frequency components and gradually recovering more on high-frequency components, but DPS$_{\boldsymbol{y}_t}$ maintains larger attention on the low-frequency recovery throughout all timesteps. This difference arises from the discrepancy between $\boldsymbol{y}$ and $\boldsymbol{y}_t$ and leads to their contrast in $\epsilon$-prediction and reconstruction error. From timestep 1000 to 600 (around) as the early stages of diffusion posterior sampling, DPS$_{\boldsymbol{y}_t}$ prohibits a smaller $\epsilon$-prediction error than DPS (the red curve of Fig.1b). Consistently accompanying with it, the reconstruction error of DPS$_{\boldsymbol{y}_t}$ is smaller and converges faster before $t = 600$ by comparing DPS (purple) and DPS$_{\boldsymbol{y}_t}$ (red) in Fig.1c.

Such observations verify our claim that the gradient of log likelihood $\nabla_{\boldsymbol{x}_t} \log p\left(\boldsymbol{y} \mid \hat{\boldsymbol{x}}_0\right)$ involved in DPS introduces more high-frequency signals during early stages which poses negative effects on early time low-frequency recovery reflecting from the larger posterior sampling error shown in DPS. The success of DPS$_{\boldsymbol{y}_t}$ in early stages arises from the similar noise and frequency level between $\boldsymbol{y}_t$ and $\hat{\boldsymbol{x}}_0$ which brings less biased guidance for low-frequency generations before $t = 600$. However, from $t = 600$ to $0$, when the diffusion model gradually transfers it focus from low-frequency to high-frequency restoration shown in Fig.1a, the random noise $\boldsymbol{u}$ in $\boldsymbol{y}_t$ brings guidance mismatching the real details, which leads DPS$_{\boldsymbol{y}_t}$ with larger posterior sampling error than DPS shown in Fig.1b and 1c. Further, in Figure 2, DPS$_{\boldsymbol{y}_t}$'s generation lacks details and looks ambiguous compared with DPS.

The proposed DPS-CM preserves the advantage of DPS$_{\boldsymbol{y}_t}$ at the early stages of sampling but still keeps accurate high-frequency generations during the later period. DPS-CM leverages the crafted measurement $\mathbf{y}_t$ instead of random sampled $\boldsymbol{y}_t$ to perform preciser posterior sampling from $\nabla_{\boldsymbol{x}_t} \log p\left(\boldsymbol{x}_t \mid \mathbf{y}_t\right)$, while $\mathbf{y}_t$ is sampled from another diffusion denoising process with the posterior $p\left(\mathbf{y}_t \mid \boldsymbol{y}\right)$. We will formulate DPS-CM in detail and discuss its advantages in Section 3.2.

### 3.2 ENHANCING POSTERIOR SAMPLING BY INTEGRATING CRAFTED NOISY MEASUREMENTS

DPS$_{\boldsymbol{y}_t}$ introduced in Section 3.1 fits the diffusion model $\epsilon_\theta(\cdot, t)$ adaptively and produces smaller $\epsilon$-prediction error, but fails to recover high-frequency components (details, domain-dependent signals) during the late denoising stage. Besides the random noise obstructing the generation of real details, this failure also origins from the intractable likelihood term $p\left(\boldsymbol{y}_t \mid \boldsymbol{x}_t\right)$ in Eq.7:

$$\nabla_{\boldsymbol{x}_t} \log p\left(\boldsymbol{x}_t \mid \boldsymbol{y}_t\right) = \nabla_{\boldsymbol{x}_t} \log p\left(\boldsymbol{x}_t\right) + \nabla_{\boldsymbol{x}_t} \log p\left(\boldsymbol{y}_t \mid \boldsymbol{x}_t\right). \tag{7}$$

Apparently, there exists no explicit dependency between the noisy measurement $\boldsymbol{y}_t$ and $\boldsymbol{x}_t$ nor $\hat{\boldsymbol{x}}_0$ applying Eq.4 given the measurement model. Thus, $\nabla_{\boldsymbol{x}_t} \log p\left(\boldsymbol{y}_t \mid \hat{\boldsymbol{x}}_0\right)$ is an inadequate estimate for $\nabla_{\boldsymbol{x}_t} \log p\left(\boldsymbol{x}_t \mid \boldsymbol{y}_t\right)$. In order to leverage the advantage of DPS$_{\boldsymbol{y}_t}$ while exploiting the tractable measurement model $p\left(\boldsymbol{y}_0 \mid \boldsymbol{x}_0\right)$ into approximation, the gradient of log posterior in DPS$_{\boldsymbol{y}_t}$ can be factorized as:

$$\nabla_{\boldsymbol{x}_t} \log p\left(\boldsymbol{x}_t \mid \boldsymbol{y}_t\right) = \nabla_{\boldsymbol{x}_t} \log \int p\left(\boldsymbol{x}_t \mid \boldsymbol{y}_0, \boldsymbol{y}_t\right) p\left(\boldsymbol{y}_0 \mid \boldsymbol{y}_t\right) d\boldsymbol{y}_0$$

$$= \nabla_{\boldsymbol{x}_t} \log \int p\left(\boldsymbol{x}_t \mid \boldsymbol{y}_0\right) p\left(\boldsymbol{y}_0 \mid \boldsymbol{y}_t\right) d\boldsymbol{y}_0$$

$$= \nabla_{\boldsymbol{x}_t} \log \mathbb{E}_{\boldsymbol{y}_0 \sim p(\boldsymbol{y}_0 \mid \boldsymbol{y}_t)} \left[p\left(\boldsymbol{x}_t \mid \boldsymbol{y}_0\right)\right]$$

$$= \nabla_{\boldsymbol{x}_t} \log p\left(\boldsymbol{x}_t\right) + \nabla_{\boldsymbol{x}_t} \log \mathbb{E}_{\boldsymbol{y}_0 \sim p(\boldsymbol{y}_0 \mid \boldsymbol{y}_t)} \left[p\left(\boldsymbol{y}_0 \mid \boldsymbol{x}_t\right)\right].$$

With the similar factorization on the posterior $p\left(\boldsymbol{y}_0 \mid \boldsymbol{x}_t\right)$, we have:

$$
\begin{aligned}
\nabla_{\boldsymbol{x}_t} \log p\left(\boldsymbol{x}_t \mid \boldsymbol{y}_t\right) &= \nabla_{\boldsymbol{x}_t} \log p\left(\boldsymbol{x}_t\right) + \nabla_{\boldsymbol{x}_t} \log \mathbb{E}_{\boldsymbol{y}_0 \sim p(\boldsymbol{y}_0 \mid \boldsymbol{y}_t)}\left[p\left(\boldsymbol{y}_0 \mid \boldsymbol{x}_t\right)\right] \\
&= \nabla_{\boldsymbol{x}_t} \log p\left(\boldsymbol{x}_t\right) + \nabla_{\boldsymbol{x}_t} \log \mathbb{E}_{\boldsymbol{y}_0 \sim p(\boldsymbol{y}_0 \mid \boldsymbol{y}_t)}\left[\mathbb{E}_{\boldsymbol{x}_0 \sim p(\boldsymbol{x}_0 \mid \boldsymbol{x}_t)}\left[p\left(\boldsymbol{y}_0 \mid \boldsymbol{x}_0\right)\right]\right] .
\end{aligned}
\tag{8}
$$

The gradient of log likelihood $\nabla_{\boldsymbol{x}_t} \log p\left(\boldsymbol{y}_t \mid \boldsymbol{x}_t\right)$ represented as the second term in Eq.8 enables us to exploit the tractable measurement model $p\left(\boldsymbol{y}_0 \mid \boldsymbol{x}_0\right)$ for the target posterior sampling. Applying the posterior mean approximation of $p\left(\boldsymbol{x}_0 \mid \boldsymbol{x}_t\right)$ in Eq.4, we can have $\hat{\boldsymbol{x}}_0$ in the estimate of $\nabla_{\boldsymbol{x}_t} \log p\left(\boldsymbol{y}_t \mid \boldsymbol{x}_t\right)$ as:

$$
\begin{aligned}
\nabla_{\boldsymbol{x}_t} \log p\left(\boldsymbol{y}_t \mid \boldsymbol{x}_t\right) &= \nabla_{\boldsymbol{x}_t} \log \mathbb{E}_{\boldsymbol{y}_0 \sim p(\boldsymbol{y}_0 \mid \boldsymbol{y}_t)}\left[\mathbb{E}_{\boldsymbol{x}_0 \sim p(\boldsymbol{x}_0 \mid \boldsymbol{x}_t)}\left[p\left(\boldsymbol{y}_0 \mid \boldsymbol{x}_0\right)\right]\right] \\
&\simeq \nabla_{\boldsymbol{x}_t} \log \mathbb{E}_{\boldsymbol{y}_0 \sim p(\boldsymbol{y}_0 \mid \boldsymbol{y}_t)}\left[p\left(\boldsymbol{y}_0 \mid \hat{\boldsymbol{x}}_0\right)\right] .
\end{aligned}
\tag{9}
$$

The approximated mean of the posterior $p\left(\boldsymbol{y}_0 \mid \boldsymbol{y}_t\right)$ is left here because the crucial selection affects the generation quality. Here, $\text{DPS}_{\boldsymbol{y}_t}$ treats the random sampled $\boldsymbol{y}_t$ as the noisy approximated mean for $p\left(\boldsymbol{y}_0 \mid \boldsymbol{y}_t\right)$ at $t$. For the case of DPS, applying $\boldsymbol{y} = \mathbb{E}\left[\boldsymbol{y}_0 \mid \boldsymbol{y}_t\right]$ makes the same gradient $\nabla_{\boldsymbol{x}_t} \log p\left(\boldsymbol{x}_t \mid \boldsymbol{y}_t\right) = \nabla_{\boldsymbol{x}_t} \log p\left(\boldsymbol{x}_t \mid \boldsymbol{y}\right)$. According to the discussion in Section.3.1, both cases are flawful: DPS increases posterior errors during early stages and $\text{DPS}_{\boldsymbol{y}_t}$ produces biased high-frequency recovery in the later period. To retain their benefits while eliminating the drawbacks, we integrate crafted measurements $\mathbf{y}_t$ into posterior sampling as follows.

**Integrate crafted measurements $\mathbf{y}_t$.** Since Eq.9 gives us a closer estimate of $\nabla_{\boldsymbol{x}_t} \log p\left(\boldsymbol{y}_t \mid \boldsymbol{x}_t\right)$, the random sampled $\boldsymbol{y}_t$ within the $\text{DPS}_{\boldsymbol{y}_t}$'s likelihood approximation $p\left(\boldsymbol{y}_t \mid \hat{\boldsymbol{x}}_0\right)$ is a rough estimate for the intractable $p\left(\boldsymbol{y}_0 \mid \boldsymbol{y}_t\right)$ and induces random noise that impacts negatively on the crisp reconstructions during the late stages. To form a tractable posterior approximation while keeping the advantage of $\text{DPS}_{\boldsymbol{y}_t}$ discussed in Section 3.1, we introduce another DDPM reverse trajectory as $\{\mathbf{y}_t\}_{t=T}^0$ along with the restoration trajectory of $\boldsymbol{x}_t$. This crafted measurement trajectory $\{\mathbf{y}_t\}_{t=T}^0$ is constructed by the intermediate generation $\mathbf{y}_t$ of a diffusion posterior sampling process with the posterior distribution $p\left(\mathbf{y}_t \mid \boldsymbol{y}\right)$. In the case of Gaussian noise, $\{\mathbf{y}_t\}_{t=T}^0$ is formed as **line 3-6** iteratively in Algorithm 1. Throughout the trajectory, $\mathbf{y}_t$ is gradually remedied to the vanilla measurement $y$. Besides, $\{\mathbf{y}_t\}_{t=T}^0$ shares the same diffusion model $\boldsymbol{s}_\theta(\cdot, t)$ with restoration trajectory $\{\boldsymbol{x}_t\}_{t=T}^0$ because $\boldsymbol{x}$ and $\boldsymbol{y}$ lie on close manifolds in most typical inverse problems. We incorporate $\mathbf{y}_t$ into the posterior distribution $p\left(\boldsymbol{x}_t \mid \mathbf{y}_t\right)$ to form DPS-CM. Unlike the random sampled $\boldsymbol{y}_t$ in $\text{DPS}_{\boldsymbol{y}_t}$, the gradually denoised crafted measurement $\mathbf{y}_t$ will not cause information loss but inject signals adaptively from low-frequency to high-frequency matching the restoration denoising trajectory of $\boldsymbol{x}$ with the close model distribution. Thus, we can estimate the posterior mean $\mathbb{E}\left[\boldsymbol{y}_0 \mid \mathbf{y}_t\right]$ of $\boldsymbol{y}_0$ in Eq.9 as $\hat{\mathbf{y}}_0$ by applying Eq.4. Replacing $\boldsymbol{y}_t$ with $\mathbf{y}_t$, the log likelihood gradient $\nabla_{\boldsymbol{x}_t} \log p\left(\mathbf{y}_t \mid \boldsymbol{x}_t\right)$ is estimated as:

$$
\begin{aligned}
\nabla_{\boldsymbol{x}_t} \log p\left(\mathbf{y}_t \mid \boldsymbol{x}_t\right) &\simeq \nabla_{\boldsymbol{x}_t} \log \mathbb{E}_{\mathbf{y}_0 \sim p(\mathbf{y}_0 \mid \mathbf{y}_t)}\left[p\left(\boldsymbol{y}_0 \mid \hat{\boldsymbol{x}}_0\right)\right] \\
&\simeq \nabla_{\boldsymbol{x}_t} \log p\left(\hat{\mathbf{y}}_0 \mid \hat{\boldsymbol{x}}_0\right) .
\end{aligned}
$$

Given this tractable approximation $\nabla_{\boldsymbol{x}_t} \log p\left(\hat{\mathbf{y}}_0 \mid \hat{\boldsymbol{x}}_0\right)$, leveraging the measurement model, it is clear to construct $\nabla_{\boldsymbol{x}_t} \log p\left(\boldsymbol{x}_t \mid \mathbf{y}_t\right)$ in the case of Gaussian noise as follows:

$$
\nabla_{\boldsymbol{x}_t} \log p_t\left(\boldsymbol{x}_t \mid \mathbf{y}_t\right) \simeq \boldsymbol{s}_\theta\left(\boldsymbol{x}_t, t\right) - \zeta_t \nabla_{\boldsymbol{x}_t}\left\|\hat{\mathbf{y}}_0 - \mathcal{A}\left(\hat{\boldsymbol{x}}_0\right)\right\|_2^2 .
\tag{10}
$$

We perform the posterior sampling incorporating the estimate in Eq.10 with crafted measurements on Gaussian deblurring and visualize the frequency pattern and sampling error in Fig.1. Shown in Fig.1a, DPS-CM (blue) concentrates more on the low-frequency recovery in the early stages compared with DPS and $\text{DPS}_{\boldsymbol{y}_t}$ and adjusts itself to high-frequency components better than $\text{DPS}_{\boldsymbol{y}_t}$ during the late period. Besides, in Fig.1b and 1c, DPS-CM (blue) has the overall smallest $\epsilon$-prediction error and achieves the best final generation closest to ground truth with the smallest reconstruction error. Empirically, combining DPS with DPS-CM leads to even better results and benefits the sampling to jump out of the plateau in the early stages of sampling. Thus, we integrate the approximated gradient of log likelihood $\nabla_{\boldsymbol{x}_t} \log p\left(\boldsymbol{y} \mid \hat{\boldsymbol{x}}_0\right)$ of DPS with our proposed $\nabla_{\boldsymbol{x}_t} \log p\left(\hat{\mathbf{y}}_0 \mid \hat{\boldsymbol{x}}_0\right)$ to facilitate the inverse problem solving. The final diffusion reverse sampling of $\boldsymbol{x}$'s restoration in DPS-CM is shown as **line 7-10** in Algorithm 1. In Appendix C, DPS-CM for measurements with Poisson noise is shown in Algorithm 2.

---

**Algorithm 1** DPS-CM

---

**Require:** Forward operator $\mathcal{A}(\cdot)$, $T$, Measurement $\boldsymbol{y}$, Step size for $\mathbf{y}$: $\{\omega_t\}_{t=1}^T$, Step size for $\boldsymbol{x}$: $\{\zeta_t\}_{t=1}^T$, Hyperparameter $\mu \in [0, 1]$ for integration control, Score function $\boldsymbol{s}_\theta(\cdot, t)$
1: $\mathbf{y}_T \sim \mathcal{N}(\mathbf{0}, \boldsymbol{I})$, $\boldsymbol{x}_T \sim \mathcal{N}(\mathbf{0}, \boldsymbol{I})$
2: **for** $t = T - 1$ to $1$ **do**

3: $\quad \boldsymbol{s}_\mathbf{y} \leftarrow \boldsymbol{s}_\theta(\mathbf{y}_t, t)$ $\qquad\qquad\qquad\qquad\qquad\qquad\qquad\qquad$ ▷ Start of $\mathbf{y}$ trajectory
4: $\quad \hat{\mathbf{y}}_0 \leftarrow \frac{1}{\sqrt{\bar{\alpha}_t}}\left(\mathbf{y}_t + (1 - \bar{\alpha}_t)\,\boldsymbol{s}_\mathbf{y}\right)$ $\qquad\qquad\qquad\qquad$ ▷ Mean Estimate of $\mathbb{E}\left[\mathbf{y}_0 \mid \mathbf{y}_t\right]$
5: $\quad \mathbf{y}'_{t-1} \leftarrow \frac{1}{\sqrt{1-\beta_t}}\left(\mathbf{y}_t + \beta_t \boldsymbol{s}_\mathbf{y}\right) + \sigma_t z_\mathbf{y}$ $\qquad\qquad$ ▷ Unconditional DDPM Sampling of $\mathbf{y}$
6: $\quad \mathbf{y}_{t-1} \leftarrow \mathbf{y}'_{t-1} - \omega_t \nabla_{\mathbf{y}_t} \|\hat{\mathbf{y}}_0 - \boldsymbol{y}\|_2$ $\qquad\qquad\qquad$ ▷ Reconstruction Loss Guidance

7: $\quad \boldsymbol{s}_{\boldsymbol{x}} \leftarrow \boldsymbol{s}_\theta(\boldsymbol{x}_t, t)$ $\qquad\qquad\qquad\qquad\qquad\qquad\qquad\qquad$ ▷ Start of $\boldsymbol{x}$ trajectory
8: $\quad \hat{\boldsymbol{x}}_0 \leftarrow \frac{1}{\sqrt{\bar{\alpha}_t}}\left(\boldsymbol{x}_t + (1 - \bar{\alpha}_t)\,\boldsymbol{s}_{\boldsymbol{x}}\right)$ $\qquad\qquad\qquad\qquad$ ▷ Mean Estimate of $\mathbb{E}\left[\boldsymbol{x}_0 \mid \boldsymbol{x}_t\right]$
9: $\quad \boldsymbol{x}'_{t-1} \leftarrow \frac{1}{\sqrt{1-\beta_t}}\left(\boldsymbol{x}_t + \beta_t \boldsymbol{s}_{\boldsymbol{x}}\right) + \sigma_t z_{\boldsymbol{x}}$ $\qquad\qquad$ ▷ Unconditional DDPM Sampling of $\boldsymbol{x}$
10: $\quad \boldsymbol{x}_{t-1} \leftarrow \boldsymbol{x}'_{t-1} - \zeta_t \nabla_{\boldsymbol{x}_t}(\mu\|\hat{\mathbf{y}}_0 - \mathcal{A}(\hat{\boldsymbol{x}}_0)\|_2 + (1-\mu)\|\boldsymbol{y} - \mathcal{A}(\hat{\boldsymbol{x}}_0)\|_2)$ ▷ Integrate Crafted $\hat{\mathbf{y}}_0$
11: **return** $\mathbf{x}_0$

---

# 4 EXPERIMENTS

## 4.1 EXPERIMENTAL SETTINGS

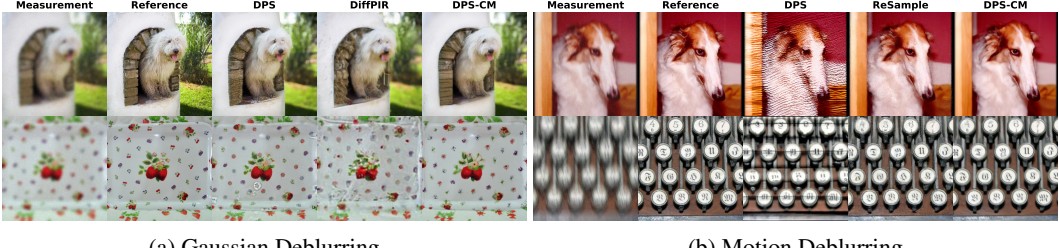

(a) Gaussian Deblurring $\qquad\qquad\qquad\qquad\qquad$ (b) Motion Deblurring

Figure 3: **Qualitative results of deblurring tasks on ImageNet dataset.** All measurements are corrupted by additional Gaussian noise with a variance of $\sigma_{\boldsymbol{y}} = 0.05$.

**Evaluation datasets and baselines.** We evaluate our proposed posterior sampling method DPS-CM on the validation set of FFHQ 256×256 and ImageNet 256×256 datasets applying DDPM sampling with 1000 timesteps. For the diffusion prior, the pre-trained diffusion models for FFHQ and ImageNet are taken from DPS (Chung et al., 2022a) and ablated diffusion model (ADM) (Dhariwal & Nichol, 2021) respectively. Detailed hyperparameter $\{\zeta_t, \omega_t, \mu\}$ settings of DPS-CM are shown in Appendix A.1. We compare our method with several recent state-of-the-art approaches including DPS (Chung et al., 2022a), Denoising Diffusion Restoration Models (DDRM) (Kawar et al., 2022), DiffPIR (Zhu et al., 2023), Optimal Posterior Covariance (OPC) (Peng et al., 2024), and also latent diffusion-based methods: Posterior Sampling with Latent Diffusion (PSLD) (Rout et al., 2024) and ReSample (Song et al., 2023a). We also compare DPS-CM with FPS-SMC (Dou & Song, 2023) and LGD-MC (Song et al., 2023b) for ablation study in Section.4.3 as they also improve posterior sampling with augmented $\boldsymbol{y}_t$ or $\boldsymbol{x}$. Although FPS-SMC relies on the separable Gaussian kernel for deblurring and LGD-MC is applied for more general tasks, they are still worthy of comparison in investigating the effect from different posterior estimate designs. We utilize the same pre-trained models used in DPS-CM for baselines DPS, DDRM, OPC, DiffPIR, FPS-SMC, and LGD-MC. For PSLD, Stable Diffusion v-1.5 (Rombach et al., 2022) is applied. For ReSample, we use the VQ-4 autoencoder and the FFHQ-LDM with CelebA-LDM from latent diffusion (Rombach et al., 2022). For OPC, we report the performance of the convert posterior covariance version with Type I guidance, as this version has the best and most stable performance among all variants. We assume that all the measurements are injected in Gaussian noise with standard deviation $\sigma = 0.05$. For mea-

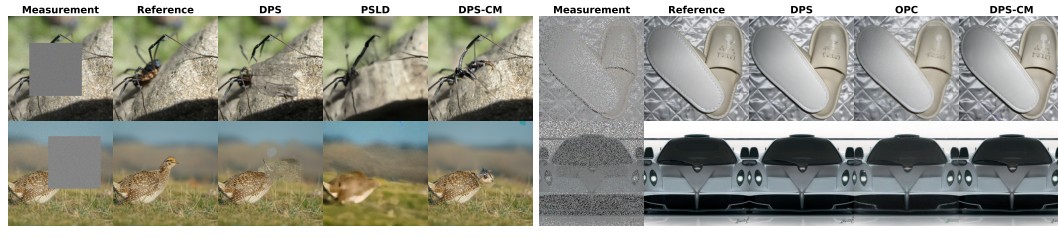

| (a) Box Inpainting | (b) Random Inpainting |

Figure 4: **Qualitative results of inpainting tasks on ImageNet dataset.** All measurements are corrupted by additional Gaussian noise with a variance of $\sigma_y = 0.05$.

surements with Poisson noise, the noise level is $\lambda = 1.0$. We run our experiments on a single GPU Nvidia A6000. We list the baseline settings in detail in Appendix A.2.

**Experimental setting for forward operators.** The forward measurement operators are mostly following Chung et al. (2022a): (1) for random inpainting, 70 percent of image pixels are masked in all channels; (2) for box inpainting, a box region with size 128×128 in an image is masked; (3) for Gaussian blur, we use the same operator implemented in DPS (Chung et al., 2022a) with kernel size 61×61 and standard deviation value 3.0; (4) We take the implementation from LeviBorodenko (2019) to form motion blur operator with kernel size 61×61 and intensity value 0.5; (5) For 4× super-resolution, we consider bicubic downsampling method; (6) For nonlinear deblurring, we utilize the approximated forward model in Tran et al. (2021) to simulate the real-world blur.

**Evaluation metrics.** We report the average performance of 100 validation samples of 4 metrics: peak signal-to-noise ratio (PSNR), structural similarity index (SSIM), Learned Perceptual Patch Similarity (LPIPS) (Dosovitskiy & Brox, 2016), and Fréchet Inception Distance (FID) (Heusel et al., 2017) for comprehensive evaluations.

## 4.2 QUANTITATIVE RESULTS

The quantitative results of DPS-CM and baselines on these noisy linear and nonlinear inverse problems are shown in Table 1, 2 and Table 3. Besides, the performance of DPS-CM for measurements with Poisson noise is shown in Appendix C. We can observe that DPS-CM can achieve the overall best performance over four metrics when solving deblurring and super-resolution problems and significantly outperform baselines on random/box inpainting problems. While the performance and stability of DPS will significantly drop when performing Gaussian/Motion deblurring on the ImageNet dataset, DPS-CM with the crafted measurements can instead form more precise and robust posterior sampling reflected from the consistent performance over these two datasets. Optimal Posterior Covariance (OPC), an improved posterior estimate by constructing the optimal posterior covariance instead of improving over the expectation, is no better than DPS-CM but still achieves comparable performance with our method. For latent diffusion-based baselines PSLD and Resample, It is interesting that they can handle the more challenging inverse problems on the ImageNet better than the FFHQ dataset, which exhibits the advantage of latent diffusion capturing on more complex visual distributions and semantic patterns. Remarkably, our method exceeds baselines with different diffusion priors when solving motion deblurring problems, shown in examples of Figure 3b. Besides, DPS-CM outperforms DPS in solving ill-posed problems, such as nonlinear blurring and inverse problems with Poisson noise as shown in Table 3 and Appendix C. In Fig.3, 4 and 5a, compared with baselines on different noisy linear inverse problems, DPS-CM shows high-quality reconstructions with mild distortion and precise detail recovery. For example, DPS-CM has better letter recovery on the keyboards in Fig.3b and signature recovery (under the car) in Fig.4b. In super-resolution and box inpainting, DPS-CM displays fewer missing details and provides reconstructions suited to the natural background and masked objects. For nonlinear deblurring in Fig.5b, DPS reconstructs measurements to the target to some extent but with extra noise and distortions injected, and DPS-CM can capture the realistic details close to the target in this nonlinear task. Additional visual examples of DPS-CM and baselines on different inverse problems with Gaussian/Poisson noise are shown in Appendix E.

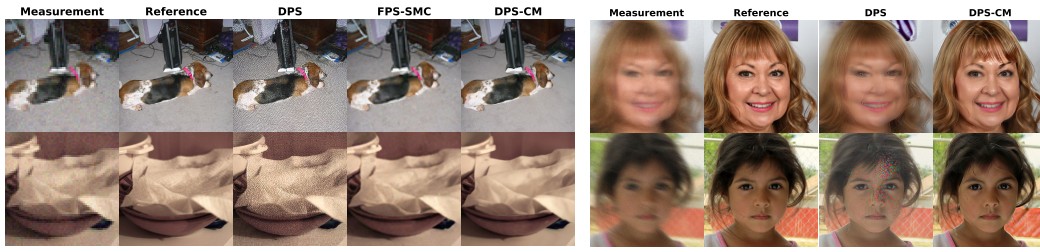

| (a) Super-resolution | (b) Nonlinear Deblurring |

Figure 5: **Qualitative results of super-resolution on ImageNet and nonlinear deblurring on FFHQ.** All measurements are corrupted by additional Gaussian noise with a variance of $\sigma_{\boldsymbol{y}} = 0.05$.

| Dataset | Method | Gaussian Deblur | | | | Motion Deblur | | | | super-resolution ($4\times$) | | | |
|---|---|---|---|---|---|---|---|---|---|---|---|---|---|
| | | PSNR ↑ | SSIM ↑ | LPIPS ↓ | FID ↓ | PSNR ↑ | SSIM ↑ | LPIPS ↓ | FID ↓ | PSNR ↑ | SSIM ↑ | LPIPS ↓ | FID ↓ |
| FFHQ | DPS-CM (*Ours*) | 27.45 | **0.7819** | **0.2059** | **56.04** | **30.31** | **0.8585** | **0.1609** | **42.58** | 27.81 | **0.7942** | **0.2008** | **51.72** |
| | DDRM | 25.20 | 0.7289 | 0.2976 | 105.06 | - | - | - | - | 26.89 | 0.7821 | 0.2667 | 95.23 |
| | DPS | 26.35 | 0.7533 | 0.2178 | 65.20 | 27.84 | 0.8023 | 0.2129 | 70.10 | 27.55 | 0.7886 | 0.2077 | 57.20 |
| | OPC | 27.06 | 0.7651 | 0.2196 | 61.76 | 26.20 | 0.7371 | 0.2535 | 66.63 | 26.90 | 0.7637 | 0.2400 | 76.33 |
| | DiffPIR | 24.13 | 0.6649 | 0.2919 | 76.96 | 26.35 | 0.7394 | 0.2487 | 64.25 | 25.74 | 0.7143 | 0.2711 | 67.19 |
| | PSLD | **28.45** | 0.7689 | 0.3019 | 99.90 | 26.19 | 0.6779 | 0.3667 | 137.88 | **27.91** | 0.7783 | 0.2783 | 87.82 |
| | ReSample | 26.10 | 0.6441 | 0.3361 | 100.21 | 28.26 | 0.7221 | 0.2937 | 89.80 | 23.23 | 0.4517 | 0.5061 | 151.20 |
| ImageNet | DPS-CM (*Ours*) | 22.73 | **0.6147** | **0.3397** | **128.92** | **24.36** | **0.6814** | **0.3163** | **97.54** | 23.78 | **0.6450** | **0.3242** | **97.38** |
| | DDRM | 21.48 | 0.5527 | 0.4948 | 242.38 | - | - | - | - | 22.72 | 0.6225 | 0.4252 | 196.32 |
| | DPS | 16.36 | 0.3449 | 0.5041 | 208.49 | 17.41 | 0.3970 | 0.4920 | 204.27 | 21.17 | 0.5404 | 0.3668 | 106.71 |
| | OPC | 18.93 | 0.4254 | 0.4579 | 132.38 | 18.43 | 0.3782 | 0.4989 | 174.44 | 19.44 | 0.4113 | 0.4836 | 169.71 |
| | DiffPIR | 20.62 | 0.4753 | 0.4445 | 154.12 | 23.38 | 0.6285 | 0.3711 | 120.58 | 22.97 | 0.5985 | 0.3840 | 119.66 |
| | PSLD | **23.45** | 0.6089 | 0.3419 | 131.90 | 23.18 | 0.5688 | 0.4206 | 167.57 | **23.79** | 0.6371 | 0.3346 | 120.68 |
| | ReSample | 22.79 | 0.5147 | 0.4355 | 172.17 | 23.95 | 0.5723 | 0.3929 | 125.33 | 21.04 | 0.3973 | 0.5001 | 203.56 |

Table 1: **Quantitative results for delur and super-resolution on FFHQ and ImageNet dataset.**
dicate the best and second best results, respectively.

## 4.3 ABLATION STUDIES

**Influence of hyperparameter $\mu$.** Here, we want to investigate the influence of hyperparameter $\mu \in [0, 1]$, which controls the proportion of proposed $\nabla_{\boldsymbol{x}_t} \log p_t (\boldsymbol{y}_t \mid \mathbf{x}_t)$ in the sampling of DPS-CM. When $\mu = 0.0$, it falls into DPS. Specifically, we can observe the performance (PSNR) of DPS-CM on box inpainting on ImageNet with different $\mu$ in Figure 6. DPS-CM achieves the best results when $\mu = 0.5$ which indicates the benefits of the interplay between $\nabla_{\boldsymbol{x}_t} \log p_t (\mathbf{y}_t \mid \mathbf{x}_t)$ and $\nabla_{\boldsymbol{x}_t} \log p_t (\boldsymbol{y} \mid \mathbf{x}_t)$. Besides, pure DPS-CM with $\mu = 1.0$ leads to a better log posterior gradient estimate and outperforms DPS-CM with $\mu = 0.0$.

Figure 6: Ablation study on $\mu$.

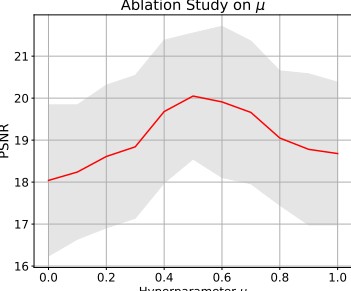

**Compare crafted measurements with other augmented posterior estimate methods.** Here, we further validate the effect of our crafted measurements $\{\mathbf{y}_t\}_{t=T}^0$ compared with DPS$_{\boldsymbol{y}_t}$, where $\boldsymbol{y}_t$ in the posterior is an i.i.d sample from the forward process $p(\boldsymbol{y}_t \mid \boldsymbol{y}_0)$ same as Eq.3, LGD-MC (Song et al., 2023b), which is a loss guided diffusion with $\boldsymbol{x}^{(i)}$ i.i.d. sampled from $p(\mathbf{x}_0 \mid \mathbf{x}_t)$ for the posterior estimate, and FPS-SMC which incorporates the $\mathbf{y}_{t-1}$ sampled from the non-parametric forward/reverse process given $\mathbf{y}$. Specifically, in the inverse problem setting, the gradient of the log likelihood LGD-MC is approximated as:

$$\mathrm{MC}_n (\mathbf{x}_t, \mathbf{y}) = \rho \nabla_{\mathbf{x}_t} \log \left( \frac{1}{n} \sum_{i=1}^n \left\| \boldsymbol{y} - \mathcal{A} \left( \hat{\boldsymbol{x}}_0^{(i)} \right) \right\|_2^2 \right),$$

| Dataset | Method | Inpainting (Random) | | | | Inpainting (Box) | | | |
|---|---|---|---|---|---|---|---|---|---|
| | | PSNR ↑ | SSIM ↑ | LPIPS ↓ | FID ↓ | PSNR ↑ | SSIM ↑ | LPIPS ↓ | FID ↓ |
| FFHQ | DPS-CM (*Ours*) | **31.20** | **0.8903** | **0.1233** | **28.05** | **23.87** | **0.8470** | **0.1331** | **35.52** |
| | DDRM | 25.49 | 0.7727 | 0.2702 | 101.26 | 20.40 | 0.8005 | 0.2265 | 65.12 |
| | DPS | 29.74 | 0.8136 | 0.1397 | 43.12 | 21.54 | 0.8127 | 0.1514 | 50.69 |
| | OPC | 29.84 | 0.8709 | 0.1615 | 50.49 | 22.69 | 0.8222 | 0.1569 | 48.01 |
| | DiffPIR | 29.40 | 0.8425 | 0.2275 | 83.09 | 21.95 | 0.7253 | 0.2484 | 63.40 |
| | PSLD | 28.52 | 0.7756 | 0.3165 | 101.04 | 21.07 | 0.6894 | 0.3761 | 128.24 |
| | ReSample | 29.65 | 0.8367 | 0.2024 | 61.44 | 19.69 | 0.7837 | 0.2452 | 136.84 |
| ImageNet | DPS-CM (*Ours*) | **26.71** | **0.8121** | **0.1684** | **37.92** | **20.05** | **0.7880** | **0.2270** | **111.01** |
| | DDRM | 21.89 | 0.6204 | 0.4162 | 230.95 | 18.32 | 0.7373 | 0.3003 | 142.95 |
| | DPS | 26.61 | 0.8060 | 0.1824 | 39.92 | 18.04 | 0.7626 | 0.2201 | 119.87 |
| | OPC | 23.46 | 0.6517 | 0.3063 | 88.47 | 16.46 | 0.6412 | 0.2910 | 168.26 |
| | DiffPIR | 25.79 | 0.7470 | 0.2873 | 106.79 | 18.08 | 0.6030 | 0.3326 | 160.95 |
| | PSLD | 25.46 | 0.6841 | 0.3509 | 104.99 | 18.17 | 0.5469 | 0.4701 | 220.96 |
| | ReSample | 25.01 | 0.7201 | 0.2825 | 98.45 | 17.50 | 0.6862 | 0.3346 | 225.58 |

Table 2: **Quantitative results for inpainting on FFHQ and ImageNet dataset.** We use **bold** and underline to indicate the best and second best results, respectively.

where $\boldsymbol{x}_0^{(i)} \sim p(\boldsymbol{x}_0 \mid \boldsymbol{x}_t)$. DPS$_{\boldsymbol{y}_t}$ with multiple Monte-Carlo samples can be formed as:

$$\nabla_{\boldsymbol{x}_t} \log p(\boldsymbol{y}_t \mid \boldsymbol{x}_t) = \rho \nabla_{\mathbf{x}_t} \log \left( \frac{1}{n} \sum_{i=1}^{n} \left\| \boldsymbol{y}_t^{(i)} - \mathcal{A}(\hat{\boldsymbol{x}}_0) \right\|_2^2 \right),$$

where $\boldsymbol{y}_t^{(i)} \sim p(\boldsymbol{y}_t \mid \boldsymbol{y}_0)$. And the estimate in FPS-SMC is $p_{\boldsymbol{\theta}}(\mathbf{y}_{t-1} \mid \mathbf{x}_{t-1})$ given sampled $\mathbf{y}_{t-1}$. We conduct experiments on $4\times$ super-resolution to demonstrate the superiority of our crafted measurements $\mathbf{y}_t$ over these augmented posterior estimate methods. The Monte Carlo sample number is set as 10 for all of them ($M = 10$ for FPS-SMC). Shown in Table 4, DPS-CM shows the best results on perception oriented metrics (LPIPS and FID) and comparable performance on the standard metrics (PSNR and SSIM) with FPS-SMC. DPS-MC is 50% more efficient than FPS-SMC as shown in the running time report in Appendix B, which validates the design in this work can keep the details and promote posterior approximation at the same time.

| Method | Nonlinear Deblur | | | |
|---|---|---|---|---|
| | PSNR ↑ | SSIM ↑ | LPIPS ↓ | FID ↓ |
| DPS-CM (*Ours*) | **22.66** | **0.6465** | **0.3626** | **122.64** |
| DPS | 21.21 | 0.6191 | 0.3846 | 136.86 |

Table 3: Quantitative results for nonlinear deblurring on FFHQ.

| Method | super-resolution ($4\times$) | | | |
|---|---|---|---|---|
| | PSNR ↑ | SSIM ↑ | LPIPS ↓ | FID ↓ |
| DPS-CM (*Ours*) | 27.81 | 0.7942 | **0.2008** | **51.72** |
| DPS$_{\boldsymbol{y}_t}$ | 22.40 | 0.6514 | 0.3273 | 96.34 |
| LGD-MC | 26.35 | 0.7674 | 0.2359 | 61.98 |
| FPS-SMC | **28.12** | **0.8053** | 0.2140 | 60.22 |

Table 4: Ablation study results on FFHQ comparing crafted measurements in DPS-CM with other augmented posterior estimate methods.

## 5 CONCLUSION

This work introduces Diffusion Posterior Sampling with Crafted Measurements to solve general and noisy inverse problems. DPS-CM leverages an additional diffusion reverse process to form a crafted measurement trajectory $\{\mathbf{y}_t\}_{t=T}^{0}$ with extensive noise levels to perform a frequency-adaptive and less biased diffusion posterior sampling with the log posterior gradient $\nabla_{\boldsymbol{x}_t} \log p(\boldsymbol{x}_t \mid \mathbf{y}_t)$ incorporating $\mathbf{y}_t$. Via experimental results, we show that our DPS-CM can generate improved high-quality restorations compared with recent state-of-the-art and also latent diffusion approaches. For the potential future works, following the idea of DPS-CM, improved design on forming more beneficial crafted measurement, e.g., additional inverse problem-related guidance for $\mathbf{y}_t$ diffusion sampling besides the reconstruction loss guidance, is a crucial direction to explore. We discuss the limitation of DPS-CM and solutions in Appendix D.

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

# A  EXPERIMENTAL DETAILS

## A.1  IMPLEMENTATION SETTINGS OF DPS-CM

Hyperparameter $\{\zeta_t, \omega_t, \mu\}$ for DPS-CM are fixed during the 1000 DDPM denoising steps. Detailed settings in the case of Gaussian and Poisson noise are shown in Table 5 and 6 respectively.

| | Gaussian Deblur | Motion Deblur | super-resolution | Inpainting(Random) | Inpainting(Box) | Nonlinear Deblur |
|---|---|---|---|---|---|---|
| FFHQ | $\{1.8, 13.0, 0.5\}$ | $\{1.7, 11.0, 0.5\}$ | $\{2.2, 8.0, 0.5\}$ | $\{3.1, 19.0, 0.285\}$ | $\{1.7, 13.0, 0.5\}$ | $\{1.6, 10.0, 0.5\}$ |
| ImageNet | $\{1.8, 13.0, 0.7\}$ | $\{1.4, 11.0, 0.5\}$ | $\{1.7, 7.0, 0.5\}$ | $\{2.4, 19.0, 0.285\}$ | $\{1.8, 13.0, 0.5\}$ | - |

Table 5: Hyperparameter settings of DPS-CM for measurements with Gaussian noise

| | Gaussian Deblur | Motion Deblur | super-resolution |
|---|---|---|---|
| FFHQ | $\{0.5, 6.0, 0.5\}$ | $\{0.4, 6.0, 0.5\}$ | $\{0.4, 3.0, 0.5\}$ |

Table 6: Hyperparameter settings of DPS-CM for measurements with Poisson noise

## A.2  IMPLEMENTATION SETTINGS OF BASELINES

**DPS**: we follow all the default $\rho$ settings in Appendix D.1 in their paper with 1000 DDPM steps.

**DDRM**: the number of steps we used is 20. DDRM with 100 steps sometimes has better results on PSNR and SSIM but worse on LPIPS and FID. The overall performance between different step schedules is very similar. For hyperparameter $\{\eta, \eta_b\}$, we follow the default setting $\{0.85, 1\}$.

**OPC**: we compare their different variants and report the Convert version with their defined Type I guidance which has the overall best performance with 50 EDM steps.

**DiffPIR**: we follow the hyperparameter settings in Appendix B.1 with 100 timesteps.

**FPS-SMC**: particle size $M$ is 10 and the $c$ value is followed the setting in their Table 8.

**LGD-MC**: LGD-MC is performed with $n = 10$. The loss function follows the settings in Appendix B.2 with $r_t = 0.05$, loss coefficient $\lambda = 10^{-3}$ and 1000 timesteps.

**PSLD**: we follow the default hyperparameter setting $\{\eta = 1, \gamma = 0.1\}$ and sample for 1000 DDIM steps.

**ReSample**: we implement it with default settings of their paper and codebase: $\{\tau = 10^{-4}, \gamma = 40\}$, conducting latent optimization with maximum 500 iterations and sampling for 500 DDIM steps.

# B  RUNNING TIME OF DPS-CM AND BASELINES

| Methods | Running Time(in seconds) | NFEs |
|---|---|---|
| DPS-CM | 123.22 | 2000 |
| DDRM | 0.54 | 20 |
| DPS | 64.70 | 1000 |
| OPC | 9.69 | 50 |
| DiffDIR | 3.92 | 100 |
| LGD-MC | 66.45 | 1000 |
| FPS-SMC | 182.67 | 1000 |
| PSLD | 517.47 | 1000 |
| ReSample | 639.51 | 500 |

Table 7: Implementation details of DPS-CM

## C ALGORITHM: DPS-CM FOR MEASUREMENTS WITH POISSON NOISE

DPS-CM is applied in the case of the measurements with Poisson noise in Algorithm 2. Its performance on the deblurring and super-resolution is shown in Tab.8.

---

**Algorithm 2** DPS-CM (Poisson)

---

**Require:** Forward operator $\mathcal{A}(\cdot)$, $T$, Measurement $\boldsymbol{y}$, Step size for $\mathbf{y}$: $\{\omega_t\}_{t=1}^T$, Step size for $\boldsymbol{x}$: $\{\zeta_t\}_{t=1}^T$, Hyperparameter $\mu \in [0,1]$ for integration control, Score function $\boldsymbol{s}_\theta(\cdot, t)$
1: $\mathbf{y}_T \sim \mathcal{N}(\mathbf{0}, \boldsymbol{I})$, $\boldsymbol{x}_T \sim \mathcal{N}(\mathbf{0}, \boldsymbol{I})$
2: **for** $t = T - 1$ to $1$ **do**

3: $\quad \boldsymbol{s}_\mathbf{y} \leftarrow \boldsymbol{s}_\theta(\mathbf{y}_t, t)$ $\hfill \triangleright$ Start of $\mathbf{y}$ trajectory
4: $\quad \hat{\mathbf{y}}_0 \leftarrow \frac{1}{\sqrt{\bar{\alpha}_t}} (\mathbf{y}_t + (1 - \bar{\alpha}_t) \boldsymbol{s}_\mathbf{y})$ $\hfill \triangleright$ Mean Estimate of $\mathbb{E}[\mathbf{y}_0 \mid \mathbf{y}_t]$
5: $\quad \mathbf{y}'_{t-1} \leftarrow \frac{1}{\sqrt{1-\beta_t}} (\mathbf{y}_t + \beta_t \boldsymbol{s}_\mathbf{y}) + \sigma_t z_\mathbf{y}$ $\hfill \triangleright$ Unconditional DDPM Sampling of $\mathbf{y}$
6: $\quad \mathbf{y}_{t-1} \leftarrow \mathbf{y}'_{t-1} - \omega_t \nabla_{\mathbf{y}_t} \|\hat{\mathbf{y}}_0 - \boldsymbol{y}\|_\Lambda$ $\hfill \triangleright$ Reconstruction Loss Guidance

7: $\quad \boldsymbol{s}_{\boldsymbol{x}} \leftarrow \boldsymbol{s}_\theta(\boldsymbol{x}_t, t)$ $\hfill \triangleright$ Start of $\boldsymbol{x}$ trajectory
8: $\quad \hat{\boldsymbol{x}}_0 \leftarrow \frac{1}{\sqrt{\bar{\alpha}_t}} (\boldsymbol{x}_t + (1 - \bar{\alpha}_t) \boldsymbol{s}_{\boldsymbol{x}})$ $\hfill \triangleright$ Mean Estimate of $\mathbb{E}[\boldsymbol{x}_0 \mid \boldsymbol{x}_t]$
9: $\quad \boldsymbol{x}'_{t-1} \leftarrow \frac{1}{\sqrt{1-\beta_t}} (\boldsymbol{x}_t + \beta_t \boldsymbol{s}_{\boldsymbol{x}}) + \sigma_t z_{\boldsymbol{x}}$ $\hfill \triangleright$ Unconditional DDPM Sampling of $\boldsymbol{x}$
10: $\quad \boldsymbol{x}_{t-1} \leftarrow \boldsymbol{x}'_{t-1} - \zeta_t \nabla_{\boldsymbol{x}_t} (\mu \|\hat{\mathbf{y}}_0 - \mathcal{A}(\hat{\boldsymbol{x}}_0)\|_\Lambda + (1-\mu) \|\boldsymbol{y} - \mathcal{A}(\hat{\boldsymbol{x}}_0)\|_\Lambda$ $\hfill \triangleright$ Integrating Crafted $\hat{\mathbf{y}}_0$
11: **return** $\mathbf{x}_0$

---

| Dataset | Method | Gaussian Deblur | | | | Motion Deblur | | | | super-resolution ($4\times$) | | | |
|---|---|---|---|---|---|---|---|---|---|---|---|---|---|
| | | PSNR ↑ | SSIM ↑ | LPIPS ↓ | FID ↓ | PSNR ↑ | SSIM ↑ | LPIPS ↓ | FID ↓ | PSNR ↑ | SSIM ↑ | LPIPS ↓ | FID ↓ |
| FFHQ | DPS-CM (*Ours*) | **26.23** | **0.7368** | **0.2348** | **66.61** | **27.53** | **0.7687** | **0.2208** | **53.19** | **26.40** | **0.7417** | **0.2552** | **72.97** |
| | DPS | 25.22 | 0.7184 | 0.2391 | 71.82 | 27.01 | 0.7608 | 0.2213 | 58.78 | 25.17 | 0.6708 | 0.3391 | 106.53 |

Table 8: Quantitative results for deblurring and super-resolution on FFHQ with **Poisson noise.** We use **bold** for the best.

## D LIMITATIONS

As the high-quality generations of DPS-CM benefit from the interplay between two posterior sampling processes for the crafted measurement's generation and $\boldsymbol{x}$'s restoration, the running time of DPS-CM is nearly twice of DPS's sampling as shown in Tabel 7. However, DPS-CM shows better or comparable performance compared with latent diffusion based methods such as PSLD, Resample, and methods with Monte Carlo such as FPS-SMC and LGD-MC with less running time. Besides, we try to improve the efficiency of DPS-CM by setting $\mu = 0$ after $t$ during the late stages of sampling as the posterior sampling with crafted measurements shows similar $\epsilon$-prediction errors with DPS during the late stages in Fig. 1b. We set the $t$ as 400 and conduct an experiment for random inpainting on FFHQ dataset. From Table 9, the accelerated variant achieves comparable performance and has the over **20%** **efficiency improvement**. Besides, the core idea of DPS-CM is based on insightful observations in posterior sampling methods. We can potentially combine crafted measurements with a more efficient method, such as DiffPIR, to achieve high-quality reconstructions and maintain efficiency. Another concern for DPS-CM is how it deals with the inverse problems with the targets and measurements from very different modalities, such as phrase retriever. In this case, utilizing the same pre-trained diffusion model to construct the crafted measurements for DPS-CM sampling can not generate stable restorations. One solution is to train a smaller diffusion model on the measurement modality for crated measurements. It should be noted that most of the current zero-shot methods are designed for typical inverse problems in which the target and measurement lie on the close manifold and perform unstably on tasks similar to phrase retriever. Thus, fitting a small model to facilitate DPS-CM for such tasks is acceptable and reasonable.

| Running Time | Method | super-resolution ($4\times$) | | | |
| --- | --- | --- | --- | --- | --- |
| | | PSNR ↑ | SSIM ↑ | LPIPS ↓ | FID ↓ |
| 123.22 | DPS-CM | **31.20** | 0.8903 | **0.1233** | **28.05** |
| 98.14 | DPS-CM (accelerated) | 30.98 | **0.8906** | 0.1349 | 36.64 |
| 64.70 | DPS | 29.74 | 0.8136 | 0.1397 | 43.12 |

Table 9: Quantitative results for random inpainting on FFHQ with Gaussian noise of **accelerated DPS-CM**

# E  ADDITIONAL VISUAL RESULTS

Additional visual examples are shown here to to compare with baselines on different tasks with Gaussian noise ($\sigma = 0.05$) from Fig. 7 to Fig. 13 and with Poisson noise ($\lambda = 1.0$) from Fig. 14 to Fig. 16.

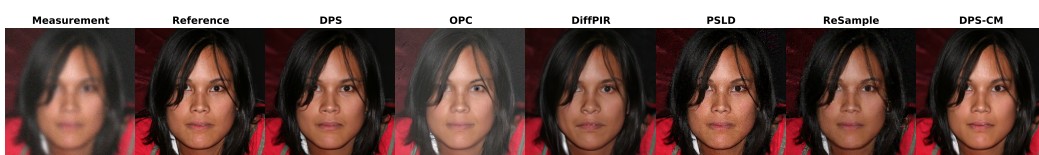

Figure 7: Additional Qualitative Results on Gaussian Deblur with Gaussian Noise.

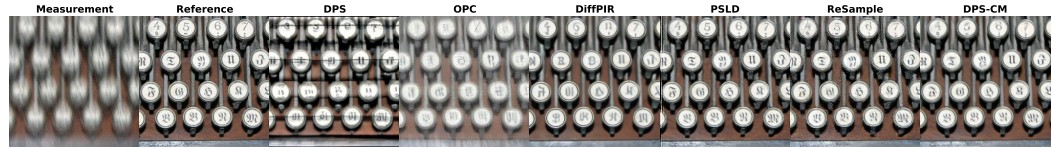

Figure 8: Additional Qualitative Results on Motion Deblurring with Gaussian Noise.

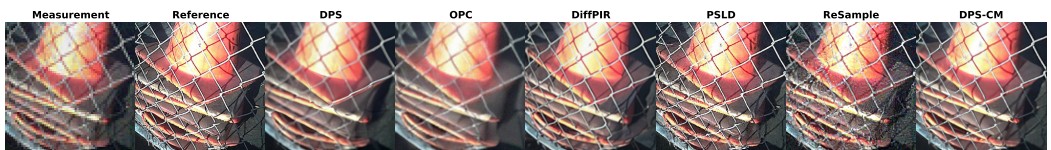

Figure 9: Additional Qualitative Results on super-resolution with Gaussian Noise.

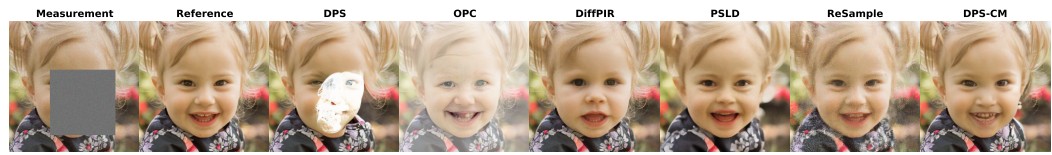

Figure 10: Additional Qualitative Results on Box Inpainting with Gaussian Noise.

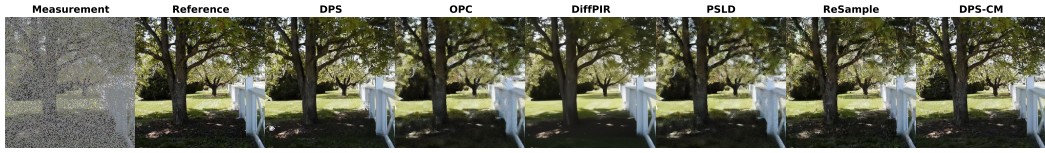

Figure 11: Additional Qualitative Results on Random Inpainting with Gaussian Noise.

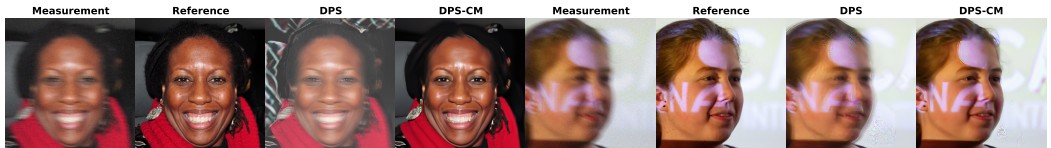

Figure 12: Additional Qualitative Results on Nonlinear Deblurring with Gaussian Noise.

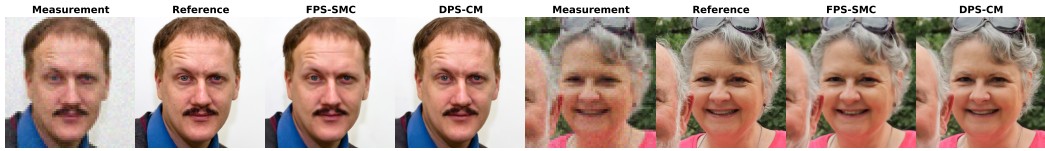

Figure 13: Additional Qualitative Results on Super-resolution with Gaussian noise compared with FPS-SMC ($M = 10$).

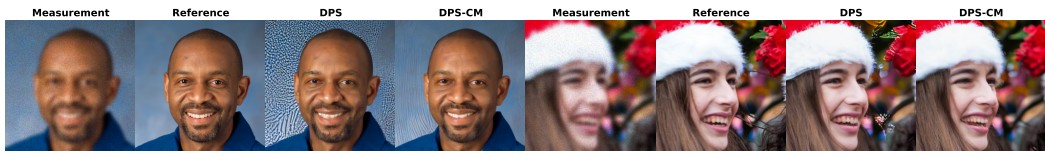

Figure 14: Additional Qualitative Results on Gaussian Deblurring with Poisson noise.

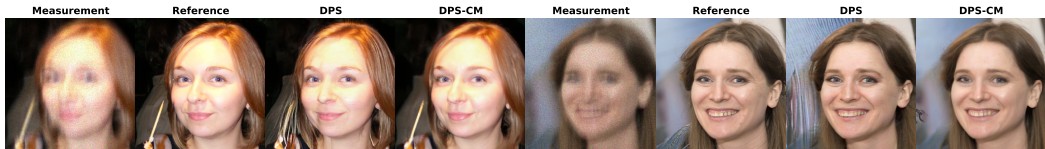

Figure 15: Additional Qualitative Results on Motion Deblurring with Poisson noise.

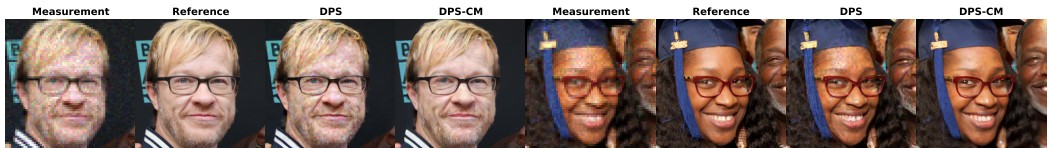

Figure 16: Additional Qualitative Results on Super-resolution with Poisson noise.

