# OpenReview forum: "Enhancing Diffusion Posterior Sampling for Inverse Problems by Integrating Crafted Measurements"
_ICLR.cc/2025/Conference — ICLR 2025 Conference Withdrawn Submission_

### Official Review · Reviewer_1Ype · 2024-10-31

**Soundness:** 3
**Presentation:** 3
**Contribution:** 1
**Rating:** 5
**Confidence:** 4

**Summary:**

The paper introduces Diffusion Posterior Sampling with Crafted Measurements (DPS-CM) for solving inverse problems, which is an improvement on DPS. DPS-CM builds on top of Diffusion Posterior Sampling (DPS) by addressing a key limitation: DPS can introduce high-frequency errors early in the sampling process when clean measurements are used. To counter this, DPS-CM replaces clean measurements with crafted noisy measurements generated through DDPM inversion. This approach aims to reduce errors by adapting the measurement's noise level to match the diffusion model's progression, improving the quality of results in general and noisy inverse problems.

**Strengths:**

1. The core contribution would be the combination of DDPM reverse trajectory and corrupted measurements as conditions of the score function.
2. The paper is well-written and easy to follow.
3. A comprehensive study of posterior sampling errors is performed.

**Weaknesses:**

1. The contribution seems incremental.
2. Intuition for designing the crafted measurements should be provided. A visualization of the measurements during the sample trajectory is also helpful for the readers to follow.
3. The method relies heavily on the crafted measurement. The result may be unstable due to different crafted measurements sampled from the reverse DDPM process.

**Questions:**

1. It seems that the method of designing crafted measurements is inspired by the gradient guidance of DPS? Have the authors tried DDIM inversion or DDPM inversion , which gradually add noise predicted by the network to the original measurements to get y_t from y?
2. It is possible to provide reconstructed results of the same image with different crafted measurements samples?

---

### Official Review · Reviewer_sDAs · 2024-11-03

**Soundness:** 1
**Presentation:** 2
**Contribution:** 1
**Rating:** 3
**Confidence:** 5

**Summary:**

This paper is about solving inverse problems with a pre-trained diffusion model. The authors claim that the injecting high frequency information early on during the diffusion process actually harms the current well known solvers. They propose to counteract this claimed shortcoming by guiding the backward process using crafted/low frequency measurements.

**Strengths:**

- The intuition and motivations are clearly explained.

**Weaknesses:**

In my opinion this paper is technically flawed, incoherent in some places and contains no review of the relevant and close literature.

- First, the presentation of the methodology starts with "We propose DPS-CM, which perform diffusion posterior sampling from $p(x_t | y_t)$ with Crafted Measurement $y_t$ belonging to another diffusion reverse trajectory $( y_t )^T _{t=0}$ instead of the vanilla input $y$". In this statement it is not clear at all what "posterior sampling" from $p(x_t | y_t )$ means. If it means running the SDE (5) with $\nabla \log p(x_t | y_t)$ then it is not justified anywhere why this would yield a valid approach for sampling from the posterior distribution $p(x|y)$ and actually, this is not the case. The explanation for the statement above is that the authors are aiming to sample from the joint distribution $p(x, y)$ and they consider the reverse diffusion that targets the same. The $y$ component of the reverse diffusion involves the score $\nabla \log p^{Y} _{t}(y_t)$ (I use the superscript $Y$ to insist that this score is **very different** from the learned, marginal, score. And the $x$ component of the diffusion uses the conditional score $\nabla \log p(x_t | y_t)$. This procedure yields a valid sample from the joint distribution $p(x,y)$ but **not** the posterior distribution $p(x|y)$ where the observation $y$ is fixed. I am thus failing to see how considering the score $\nabla \log p(x_t | y_t )$ yields a valid procedure for sampling from $p(x|y)$. As far as I am concerned this is a critical aspect that is not discussed nor explained.

- Now I turn to the score $\nabla \log p^{Y} _{t}(y_t)$. This score is very different from the pre-trained one yet the authors claim in the paper  that "Besides, $(y_t ) ^{0} _{t = T}$ shares the same diffusion model $s _{\theta}(\cdot, t)$ with restoration trajectory $(x_t)^0 _{t=T}$ because $x$ and $y$ lie on close manifolds in most typical inverse problems." In my opinion this claim deserves a lengthy discussion since it is far from obvious that this is true. For example, when the observation is an image with missing half, we can safely assume that the underlying distribution of the observation is that of the images from the dataset with missing half. Thus, running the diffusion process with the true score $\nabla \log p^{Y} _{t}(y_t)$ will return images with a missing half, whereas using the pre-trained score returns complete images. The same holds for many common inverse problems.

- The main motivation of the paper is that the injection of high frequency information early on during the diffusion process can be harmful to the reconstructions. The claim is that this can be avoided by using instead $\nabla \log p(y_t | x_t)$. Now assuming that this is true, from the derivations in the paper it is shown that one, in this case, has to estimate $\nabla \log \mathbb{E}_{y_0 \sim p(y_0 | y_t)} [p(y_0 | \hat{x}_0) ]$. In the paper this is estimated using a DPS approximation but if we assume that we can actually sample exactly from the distribution $p(y_0 | y_t)$, we would actually (and ideally) use a Monte Carlo estimate with samples $y_0 ^1, \dotsc, y_0 ^N$  from $p(y_0 | y_t)$:
$$
\nabla \log \mathbb{E} _{y_0 \sim p(y_0 | y_t)} [p(y_0 | \hat{x}_0) ] \approx \frac{1}{N} \sum _{i = 1}^N \nabla \log p(y_0 ^i | \hat{x}_0) .
$$
which is an average over "high frequency information" and is at odds with the claims of the paper. Using instead the DPS approximation does indeed involve low frequency information but this doesn't align with the theoretical justification of the paper.

- As mentioned in the paper, the "crafted measurements" are sampled from the forward process $p(y_{1:T} | y_0)$. As explained above, the way this is done in practice is **flawed**. I am failing to understand why the authors do not simply sample the observations from the forward process of the Diffusion? This also gives **exact** samples. I understand that sampling forward means that all the samples need to be stored in memory, but one can also sampled from this distribution backwards (and exactly) by noting that $p(y_{1:T} | y_0) = p(y_T |y) \prod_{t = 1} ^{t-1} p(y_t | y_0, y_{t+1})$. Also the step 10 still involves the DPS approximation. This is again at odds with the justifications provided in the paper.

- Finally, the literature review in this paper is quite poor in my opinion. There are many papers in the literature that use the noised observations $y_{1:T}$ within the backward process. A few examples are listed below. The authors compare to [3] but the similarities are never discussed.

[1] Lugmayr, Andreas, Martin Danelljan, Andres Romero, Fisher Yu, Radu Timofte, and Luc Van Gool. "Repaint: Inpainting using denoising diffusion probabilistic models." In Proceedings of the IEEE/CVF conference on computer vision and pattern recognition, pp. 11461-11471. 2022.

[2] Trippe, Brian L., Jason Yim, Doug Tischer, David Baker, Tamara Broderick, Regina Barzilay, and Tommi Jaakkola. "Diffusion probabilistic modeling of protein backbones in 3d for the motif-scaffolding problem." arXiv preprint arXiv:2206.04119 (2022).

[3] Dou, Zehao, and Yang Song. "Diffusion posterior sampling for linear inverse problem solving: A filtering perspective." In The Twelfth International Conference on Learning Representations. 2024.

**Questions:**

N/A

---

### Official Review · Reviewer_TK9L · 2024-11-05

**Soundness:** 2
**Presentation:** 3
**Contribution:** 2
**Rating:** 5
**Confidence:** 3

**Summary:**

The paper introduces DPS-CM, a new approach to improve diffusion posterior sampling (DPS) models for inverse problems like image deblurring and super-resolution. Traditional DPS methods incorporate the degraded measurement directly, leading to high-frequency errors during the early stages of the denoising process. DPS-CM addresses this by generating "crafted measurements" through an intermediate diffusion reverse process that better matches the evolving noise characteristics in the generated images. This tailored approach allows for cleaner, more accurate reconstructions, especially in high-noise scenarios. Experimental results demonstrate DPS-CM's superiority over existing methods, suggesting it as a more effective framework for leveraging diffusion models compared to vanilla DPS in various inverse problems.

**Strengths:**

1, The manuscript is well-written in most of its part and easy to follow in its motivation behind the designed crafted sampling trick. The numerical validation of this motivation to design $\nabla \log p(x_t|y_t)$ is interesting and insightful.

2, This sampling trick is easy to implement without too many changes of original DPS method.

3, The numerical validation shows the effectiveness of this proposal.

**Weaknesses:**

1, The proposed crafted measurement $y_t$ in lines 3–6 of Alg1. is based on an underlying assumption that the measurement sample lies within the domain of the deep score network $s_\theta$. However, this assumption may not hold for various inverse problems, such as CT/MR imaging and phase retrieval, where the measurement distribution often differs from that of the training images $x_0$. This discrepancy may fundamentally limit the practical contribution of the DPS-CM method, as it is unclear how effective the approach would be if the measurements differ significantly from the training data distribution.

2, The numerical comparisons provided may not accurately or fairly represent the performance of the baseline methods. For example, in Table 5 in Appendix A, hyperparameters for DPS-CM are carefully fine-tuned, whereas the baselines use their original settings. Given that the experimental setup does not strictly follow the configurations of each baseline, the reported performance gains for DPS-CM may appear more significant than they truly are.

3, The connection between lines 3-6 in Algorithm 1 and the estimation of the posterior mean $\mathbf{E}[y_0 | y_t]$ is not clearly explained. It remains unclear how the update for $y_{t-1}$ is theoretically derived. Providing additional clarification would enhance understanding of the theoretical basis for this update.

4, Although the idea of moving from “low-to-high” frequency for improved sampling is interesting, the proposed approach ultimately employs a relatively simple convex combination of the likelihood $p(y|x_t)$ with a fixed measurement $y$. This limits the contribution to a primarily engineering-oriented approach, lacking deeper theoretical connections as the paper intially suggested. Additionally, the extra hyperparameters introduced make the algorithm more challenging to fine-tune.

**Questions:**

1, From Figure 1, it appears possible to empirically track the high/low frequency ratios. This raises the question of whether $\text{DPS}_{y_t}$ could achieve similar performance with a specifically designed trade-off parameter $\mu_t$. Exploring this possibility could provide insights into optimizing the balance between high and low frequencies for improved performance.

2, What evaluation metric you are using to decide the optimal hyperparameters ?

---

### Official Review · Reviewer_9c5y · 2024-11-06

**Soundness:** 2
**Presentation:** 2
**Contribution:** 2
**Rating:** 3
**Confidence:** 4

**Summary:**

This manuscript presents a diffusion-based generative method for addressing general inverse problems in imaging. The approach involves sampling from the posterior distribution using Langevin dynamics, with an update rule consisting of two components: unconditional prior term and log-likelihood gradient term. The main contribution of the manuscript is an improved approximation for the log-likelihood gradient term, enhancing the previous approximation introduced by [1]. While [1] uses a gradient that directs toward the measurement $y$, this manuscript proposes replacing it with a convex combination of two gradients: one that guides toward the original measurement $y$, and another that guides toward a smoothed measurement $\hat{y}_0$. To obtain $\hat{y}_0$, the authors apply an additional backward diffusion process, $\{y_T, \dots, y_t, \dots, y_0\}$, which gradually transforms random Gaussian noise into the input measurement. At each time step, $\hat{y}_0$ is estimated from $y_t$ using the expectation $\hat{y}_0 = E[y_0 | y_t]$.

The proposed method is evaluated on various inpainting, super-resolution, and deblurring tasks with comparisons to competing approaches. In all presented experiments, the proposed method demonstrated improved results over [1] and other competitors.

[1] Hyungjin Chung, Jeongsol Kim, Michael T Mccann, Marc L Klasky, and Jong Chul Ye. Diffusion posterior sampling for general noisy inverse problems. arXiv preprint arXiv:2209.14687, 2022a.

**Strengths:**

The proposed algorithm is intuitively appealing, as the additional gradient that guides toward the smoothed measurement $\hat{y}_0$ aids in achieving better reconstruction of low-frequency components. The algorithm is evaluated across various experiments on two datasets using several performance metrics. In all presented experiments, the proposed method demonstrates improvements over [1].

**Weaknesses:**

Section 3 raises some questions that, I hope, the authors can help clarify.

First, could the authors clarify the difference between $y_t$ in line 291 (noisy measurements) and $\text{y}_t$ as defined in line 295 (crafted measurements)? As I understand it, noisy measurements are sampled from the forward trajectory of the $\\{y_t\\}$ diffusion process, while the crafted measurements are generated during the reverse-time trajectory. Does the direction of the diffusion impact the definition of these $y_t$ values?

Additionally, could the authors please clarify why $E[y_0 | y_t] = y$ in line 285, but $E[\text{y}_0 | \text{y}_t] = \hat{\text{y}}_0$ in lines 303-304?

I also have a concern regarding the equation in lines 306-307. Consider the scenario where, in the equation in line 126, the operator $A(\cdot)$ is identity, and noise $\eta$ is constant zero. In other words, $y = x_0$. Also, assume that $p(x_0 = 0) = 1$, so $x_0$ is a constant, specifically $x_0 = 0$. Under these conditions $y = x_0 = \hat{y}_0 = \hat{x}_0 = 0$, and consequently the right-hand side of the equation in lines 306-307 is constantly 0.

Now let us demonstrate that the left-hand side of the equation can be arbitrary far from zero.  Assume that $x_t = x_0 + n_x = n_x\ ,$ and $y_t = y_0 + n_y = n_y\ .$ Additionally, suppose that $n_x$ and $n_y$ are Gaussian-distributed with $n_x \sim N(0, 1)$ and $n_y|n_x \sim N(n_x, 1)\ .$ Then we have $p(y_t | x_t) = (2\pi)^{-0.5}\exp\\{-0.5(y_t - x_t)^2\\}$, which yields $\nabla_{x_t}\log p(y_t | x_t) = y_t - x_t$. Since both $y_t$ and $x_t$ are unbounded, the left-hand side of the equation in lines 306-307 is also unbounded, and can be arbitrary far from zero.

**Questions:**

Describing the algorithm in the introduction in plain language may increase the paper's readability. I suggest postponing the usage of symbols to the subsequent sections.

---

> ### Comment · Reviewer_9c5y · 2024-11-26
>
> After carefully reviewing the feedback from other reviewers, I have decided to maintain my original rating.

---

### Note · Authors · 2024-11-26

I have read and agree with the venue's withdrawal policy on behalf of myself and my co-authors.